# Gender and Observed Complexity in Palliative Home Care: A Prospective Multicentre Study Using the HexCom Model

**DOI:** 10.3390/ijerph182312307

**Published:** 2021-11-23

**Authors:** Xavier Busquet-Duran, Eduard Moreno-Gabriel, Eva Maria Jiménez-Zafra, Magda Tura-Poma, Olga Bosch-DelaRosa, Anna Moragas-Roca, Susana Martin-Moreno, Emilio Martínez-Losada, Silvia Crespo-Ramírez, Lola Lestón-Lado, Núria Salamero-Tura, Joana Llobera-Estrany, Ariadna Salvago-Leiracha, Ana Isabel López-García, Josep María Manresa-Domínguez, Teresa Morandi-Garde, Eda Sara Persentili-Viure, Pere Torán-Monserrat

**Affiliations:** 1Home Care Program, Granollers Support Team (PADES), Vallès Oriental Primary Care Service, Catalan Health Institute, 08520 Granollers, Spain; evajiza@gmail.com (E.M.J.-Z.); turapoma@hotmail.com (M.T.-P.); olga.bosch@creuroja.org (O.B.-D.); amoragas_roca@hotmail.com (A.M.-R.); susmartin0@hotmail.com (S.M.-M.); emlosada1974@gmail.com (E.M.-L.); silvia.crespo@creuroja.org (S.C.-R.); lestonlola@gmail.com (L.L.-L.); nsalamero.mn.ics@gencat.cat (N.S.-T.); jolloes@hotmail.com (J.L.-E.); aln1982@hotmail.com (A.S.-L.); anaisabellg@hotmail.com (A.I.L.-G.); 12450@comb.cat (T.M.-G.); eda.persentili@creuroja.org (E.S.P.-V.); 2Research Support Unit Metropolitana Nord, Primary Care Research Institute Jordi Gol (IDIAPJGol), 08303 Mataró, Spain; jmanresa@idiapjgol.info (J.M.M.-D.); ptoran.bnm.ics@gencat.cat (P.T.-M.); 3Department of Nursing, Autonomous University of Barcelona, 08193 Barcelona, Spain

**Keywords:** end-of-life, gender, palliative care, home care, complexity

## Abstract

This study analyses gender differences in the complexity observed in palliative home care through a multicentre longitudinal observational study of patients with advanced disease treated by palliative home care teams in Catalonia (Spain). We used the HexCom model, which includes six dimensions and measures three levels of complexity: high (non-modifiable situation), medium (difficult) and low. Results: *N* = 1677 people, 44% women. In contrast with men, in women, cancer was less prevalent (64.4% vs. 73.9%) (*p* < 0.001), cognitive impairment was more prevalent (34.1% vs. 26.6%; *p* = 0.001) and professional caregivers were much more common (40.3% vs. 24.3%; *p* < 0.001). Women over 80 showed less complexity in the following subareas: symptom management (41.7% vs. 51,1%; *p* = 0.011), emotional distress (24.5% vs. 32.8%; *p* = 0.015), spiritual distress (16.4% vs. 26.4%; *p* = 0.001), socio-familial distress (62.7% vs. 70.1%; *p* = 0.036) and location of death (36.0% vs. 49.6%; *p* < 0.000). Men were more complex in the subareas of “practice” OR = 1.544 (1.25–1.90 *p* = 0.000) and “transcendence” OR = 1.52 (1.16–1.98 *p* = 0.002). Observed complexity is related to male gender in people over 80 years of age. Women over the age of 80 are remarkably different from their male counterparts, showing less complexity regarding care for their physical, psycho-emotional, spiritual and socio-familial needs.

## 1. Background

There is currently a certain consensus that, beyond the vital prognosis or the disease, the backbone of palliative care should be the level of care complexity required by the person with advanced disease [1]. Complexity emerges from the interdependence between those elements involved in each case, yielding non-linearity, blurriness and chaoticity [2]. This means that complexity goes beyond the needs of independent physical, psychological, social and spiritual domains, and includes how patients interact with their families and professionals and how services respond to their needs [3,4].

The complexity and uniqueness of each patient and each situation calls for an open and multiple approach, bearing in mind that the application of conventional approaches—such as reductionism and compartmentalisation—can be potentially counterproductive [5].

To this end, a systemic understanding of these situations emphasises the ecological framework in which they unfold [6], taking into account the interactions between the people who make up the unit to be treated, and between it and their care, socioeconomic and cultural environment, as well as their evolution over time [7]. The history of a complex system evolves permanently, linked to its context and its changes [8].

Over time, androcentrism in biomedical sciences has invisibilised women, assimilating them to men [9]. To give just one example, gender differences in opiate receptors are not taken into account, and women and men are prescribed indistinctly [10]. Unfortunately, this gender bias also affects social research. As Carol Gilligan noted, Kohlberg’s foundational work on moral development drew exclusively on studying male children, yet it has strongly influenced bioethics [3,11].

Standing at the crossroads of biomedical and social research and practice, palliative care has not escaped this tendency to ignore women’s singularities. For instance, Gott et al. have recently shown that while cancer guidelines recommend an early administration of palliative care, women have shown lesser improvements in quality of life and emotional status than men [12]. Similarly, one of the most prevalent and severe symptoms in women, asthenia, is rarely registered in clinical records. In a study conducted in two Swedish hospitals, 76% of the 720 people treated in palliative care units reported having asthenia; however, only 19% of the clinical records contained this symptom [13]. Additionally, this study shows that women need to report higher levels of pain than men in order for it to be registered [13]. As a result of this emerging evidence, the status quo of considering the palliative care patients “male by default” has recently been contested [12,14]. Furthermore, this bias is compounded as it coexists with other stereotypical depictions of men and women. These exclude men from activities related to care and assume care expertise and selflessness in women [15].

Currently, there is a lack of studies on these gender dynamics in the practice of palliative home care. In this article, we explore the role of gender in the process of defining the complexity of each case. To do so, we draw on the Hexagon of Complexity (HexCom) [16,17], which defines complexity as the gap between the needs of the patient and available resources [4], registering it as a key element in the practice of palliative care [18].

## 2. Objectives

Our objectives were to analyse differences by gender in the complexity of patients registered by a home palliative care team and describe the role of age in these differences.

## 3. Methods

Design: Longitudinal observational study of a multicentre cohort.

Population: Patients requiring advanced illness and/or end-of-life treatment provided by home care support teams in Catalonia (PADES) [19].

Inclusion/exclusion criteria: Patients requiring advanced illness and/or end-of-life treatment provided by PADES teams in their own home. Patients receiving care in a residential institution were excluded. Consecutive sampling was used during the recruitment period (January 2016–December 2019) to assign the patients who met the inclusion criteria to any of the PADES teams who agreed to participate in the study.

Complexity was assessed after the demise of the patient by the interdisciplinary PADES teams using the HexCom model.

HexCom includes six dimensions (clinical, psychological, spiritual, social/family, ethical and death-related) and 18 subareas, with three levels of complexity: low, medium and high. High complexity occurs when the team cannot respond to the need and has to refer the case and/or assume that they can only act as support. Medium complexity occurs when the management is considered difficult and requires the support of other services. In cases of low complexity, the need of the patient and/or family can be easily managed (Table 1). The HexCom model presents a high internal consistency (Cronbach alpha > 0.80) and stability (test-retest: weighted kappa > 0.80), and usefulness for oncological and non-oncological patients, regardless of the stage of their condition and for all levels of care (agreement between evaluators: kappa weighted > 0.80) [20]. The apparent validity shows a high level of agreement, with a content validity index (CVI-I) over 0.92 for the model and over 0.80 for all the derived instruments [17,21]. A recently published systematic review considers the HexCom model one of the most comprehensive for assessing complexity [18].

Disease groups: Medical diagnosis (ICD-10) and advanced disease groups: cancer, advanced chronic organ failure, advanced neurological disease, dementia and geriatric frailty/multimorbidity. The PADES team had to agree on the main diagnosis associated with death.

Socio-demographic data: Age; gender; relationship with caregiver: partner, yes or no; professional caregiver: yes or no.

Patient status: Functional status (Barthel index) and mental status (Pfeiffer test).

Procedure: To standardise data collection, the 43 PADES teams (74.3% teams accepted our invitation to participate) received 10 h of face-to-face training, a user guide, and were able to call the researchers during fieldwork if necessary.

Statistical analysis: Gender (female or male) was analysed in relation to complexity, using a logistic regression analysis, adjusted for age. Each specific subarea was considered for the analysis. Except for length of care, all variables analysed were categorical and are described with their absolute and relative frequencies. To compare proportions, we used Pearson’s Chi Square test. The significance level was set at *p* ≤ 0.05. SPSS (IBM, Armonk, NY, USA) for Windows statistical package, version 25.0, was used for all analyses.

Ethics Committee Approval: The study was approved by the Clinical Research Ethics Committee of the Institute for Research in Primary Care (IDIAP) Jordi Gol (registration number P15/171) and by the Clinical Research Ethics Committees of all participating centres. All participants read and signed an informed consent form.

## 4. Results

A total of 1677 patients were recruited (44% women) (Table 2). Only 19% patients were under 65 years of age. In the group over 80 years, 52% patients were women (52%) (Table 2). In the group of women, there were fewer cancer patients (64.4% vs. 73.9%) (*p* < 0.001), more cognitive impairment (34.1% vs. 26.6%; *p* = 0.001), less partners as primary caregivers (25.2% vs. 64.5%; *p* < 0.001) and more professional caregivers (40.3% vs. 24.3%; *p* < 0.001)

As a result of statistically analysing the six dimensions of complexity as a whole, observed complexity was significantly lower in women than in men in all areas except for ethics (Table 3). The dimensions with the largest differences (around 10%) were the clinical, the spiritual and death (Figure 1). These differences in complexity occurred only in patients over the age of 80. Women over 80 years of age presented less complexity than men in the management of symptoms (41.7% vs. 51.1%; *p* = 0.011), of psycho-emotional (24.5% vs. 32.8%; *p* = 0.015), spiritual (16.4% vs. 26.4%; *p* = 0.001) and socio-familial distress (62.7% vs. 70.1%; *p* = 0.036), and in relation to the location of the death (36.0% versus 49.6%; *p* < 0.000). In the group of 80 years and under, the only subarea with a statistically significant difference was transcendence, which relates to fear of death or the future, with less complexity detected for women than for men (24.6% versus 31, 1%; *p* = 0.046). Logistic regression (Table 4) shows that people over 80 are less complex in all subareas except for the desire to accelerate death OR = 1.16 (0.79–1.71 *p* = 0.54), and that men are more complex especially in the socio-family subarea “practice” OR = 1.544 (1.25–1.90 *p* = 0.000) and the spiritual subarea “transcendence” OR = 1.52 (1.16–1.98 *p =* 0.002).

## 5. Discussion

The results of this study highlight important gender differences in observed complexity in patients from palliative home care over 80 years of age. In this group, there are more women who, compared with men, have a lower prevalence of cancer, more cognitive impairment, less partners as primary caregivers and more caregivers who are professional. The complexity observed by the professionals in this group of women was lower than in men regarding location of death, management of symptoms and psycho-emotional, spiritual and socio-familial distress.

Currently, more men than women require specialised home palliative care (the male/female proportion of people requiring palliative home care is 52% versus 48%), in agreement with the literature and with predictions of the World Health Organisation based on the different causes of death according to gender [21]. However, from the age of 80 onwards, the proportion of women increases, with more women in the group of the very elderly [22]. According to Etkind et al. [22], dementia is amongst common causes of death in women in the very elderly group and will be, alongside cancer, the main drivers of a marked increase in the demand for palliative care.

Observed complexity is higher for both men and women under 80 years of age. The inverted association between age and complexity, also observed in other studies [23], may result from a greater acceptance of death by older adults and their families. Additionally, older people with cancer tend to report less pain and negative psychosocial impact [24]. When studying patients with breast cancer, young women present more psychological distress, poorer quality of life and greater difficulties with their partners compared to older ones [25]. Importantly, the association between complexity, gender and age in relation with home palliative care agrees with recent findings showing that the proportion of hospital deaths decreases with increasing age and is higher among men than women [26], and that fewer end-of-life location changes are reported in women [27].

Finally, we should underscore that the subgroup of women over 80 years shows less complexity. In the Catalan context, this could be explained by the resilience of this subgroup, which has endured wars, post-war hardship, dictatorship, emigration and widowhood, and have been great fighters and caregivers. In addition, this could be related to different approaches to end of life. Most women have lost their partners (only 8.4% have their partner as primary caregiver versus 45.9% of men), and might have accepted that death is the next step in their biography, adopting coping strategies focused on problem solving and help seeking, whereas men draw more on avoidance and disengagement [28]. Similarly, while women actively participate in doctor–patient communication, men acquire a passive role and often refuse to talk about problems related to death [29]. Likewise, women with advanced cancer develop a more accurate understanding of their disease than men do [30]. Women sign more advance directives, reject proposed treatments more often and tend to agree more to palliative care, while men demand more continuous palliative sedation or euthanasia [31,32]. In a recent study in the Mediterranean area, Tuca et al. [33] observed a higher prevalence of ethical dilemmas in men than in women.

The higher prevalence of dementia (46.9% versus 36.5% of men) might also contribute to the lower levels of observed complexity in women. The impact of having a partner might partly explain these differences: emotional ties are crucial for older cancer patients, and male patients, unlike women, depend mainly on the social support of their partners [34]. Distress is enhanced in stressful situations when communication with the partner is dysfunctional and presents negative dyadic coping patterns [17,35,36]. Moreover, having an elderly, fragile partner can increase the feeling of burden and distress [37].

Studies indicate that men and women face the end-of-life situation differently [29]. Normative gender roles often put men in fight mode and whereas women focus on seeking help, men tend to avoid and detach themselves from the problem at hand. It has been observed that while women actively engage in doctor–patient communication, men assume a passive role and often refuse to talk about issues related to death [30]. Similarly, women with advanced cancer develop a more accurate understanding of their disease, and acknowledge the advanced stage of their condition up to 6 times more than men [30].

According to our results, spirituality is more used by women than by men as a coping mechanism for extreme situations such as death. In our study, women show less fear of death and the future. Many studies conducted in various cultural settings [4] confirm a greater degree of spirituality and religiosity in women [34,38]. A Danish study with 6640 women cancer survivors observed that women showed more spiritual, religious and existential concerns (OR 1.38) and that these were related to existential questioning and guilt [39]. Existential questioning emerges in extreme situations. Feelings of guilt might be related to normative femininity, with subordination to the judgment of others, which is closely linked to low self-esteem, the pursuit of models of perfection and, consequently, to guilt [40].

Finally, we should emphasise that no differences were found between men and women in relation to ethics, even though the literature shows that while more women sign advanced directives [32], reject more often the treatments proposed [41] and accept more readily the prescribed palliative care [31], more men request continuous palliative sedation and even euthanasia [42]. In a context like ours where euthanasia and medically assisted suicide are penalised, we anticipated more complexity in men.

### Strengths and Limitations

Gender biases and inequalities exist at multiple levels of health research and practice, affecting each other’s significance [9,43]. For instance, our analysis has not considered the gender of the primary caregiver, even though we know that three out of four informal caregivers are women [28]. Moreover, we cannot rule out a potential gender bias in the observers, since 90.8% (196/216) of the researchers were women, and women might normalise and minimise the needs of older women. Some studies point at the relevance of the gender of the evaluator [29,44], even indicating that women with heart attacks have higher mortality rates when treated by male doctors. Additionally, female doctors use a closer communicative style and female nurses spend much more time than male nurses interacting with patients [45,46,47]. Lastly, we are still constrained by gender binarism, where the gold standard is male. Other genders have not been yet taken into account, and their perspectives should be incorporated in future studies.

Intersectionality is one of the strengths of this study [14], as reflected in the multidimensional approach of the HexCom model. We avoided the unidimensional approach to understanding end-of-life circumstances, since an individual’s experiences are not shaped “by a single axis of social division, whether by race, gender, or class, but by many axes that work together and influence each other” [12]. Another strength of the study is the interdisciplinarity of our teams, which undoubtedly provides a more accurate assessment of reality than the gaze of a single observer.

We should also ask whether a change in traditional patterns is taking place. For instance, while the number of women caregivers is higher than men, a recent national survey conducted in Sweden indicated an equal number of men and women acting as informal caregivers [48]. The experienced gender inequities in home care should also be studied using qualitative methodology similarly to Sutherland’s approach [15], who asks the following straightforward question to patients, caregivers and professionals: “Do you think that the fact of being a man/woman affects your experience and, if so, how?” Finally, the role of spirituality by gender as a coping mechanism at the end of life should also be investigated.

Our results aim to raise awareness of gender inequities amongst palliative care professionals to prevent pattern perpetuation, feelings of guilt and to provide unconditional support to the “obligated caregiver”. This is one of the most extensive and comprehensive series published, which underlines the lack of resources in home palliative care derived from the inequalities and iniquities inherent of a patriarchal and neo-liberal socio-political context [15], where power discourses favour individualism and efficiency. We strongly believe in the importance of raising awareness about these differences in the context of palliative home care as a crossroads of invisibilities: women with respect to men, the dying versus the healthy, the old with respect to the young, death with respect to life, the home with respect to the hospital.

## 6. Conclusions

Our study highlights the need to address the social determinants of health with active and all-encompassing policies, as well as an intersectional viewpoint. That is, public health policies should not focus exclusively on the economic determinants, nor consider a gender perspective only and in isolation within organisational development plans or research programs.

As we have shown, especially regarding men over 80 years old, gender stereotypes increase patients’ needs and the necessary resources to address them, insofar as these cultural representations normalise lacking coping strategies and a healthier acceptance of death. Consequently, developing a “pedagogy of death and dying” with a gender perspective and at a population level would be adequate. This would be particularly urgent and timely in the Spanish context, since the passage of the euthanasia law has brought to the fore a social debate on death and dying.

Finally, more research that is inclusive is also needed, taking into account aspects such as non-binary categories and the gender of the formal and informal caregivers, who are key figures in palliative home care.

## Figures and Tables

**Figure 1 ijerph-18-12307-f001:**
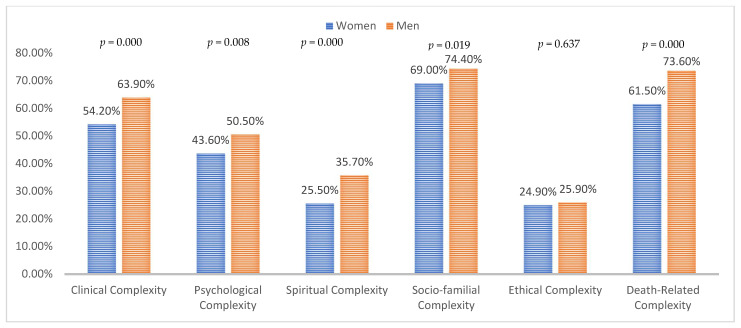
Evaluation of Complexity: Includes patients with medium or high complexity. Gender differences.

**Table 1 ijerph-18-12307-t001:** The HexCom model.

***HexCom-Clin***_2019_ **Model for the care of people with advanced disease and/or end of life situation.** Extended version for the assessment of needs and resources. OBSERVED CARE COMPLEXITY.						
**NEEDS:** Identify the patient’s areas of discomfort and relate it to the possibility of response from the service. *Mark the level of complexity of the affected areas*: ***L Low, M Medium, H High***. *Mark **N** for areas Not evaluated/Not applicable.* COMPLEXITY AND INTERPRETATION LEVELS: **L**—Low (little difficulty). Guarantees of being able to attend to the situation with the resources of the service. **M**—Medium (moderate difficulty). Guarantees of taking on the situation with the support of other professionals and/or specialised teams. **H**—High (refractory difficulty). Little chance of change. It is necessary to escort and/or probably refer to another resource/level of care.						
**Area**	**Sub-Area**		N	L	M	H
**CLINIC**	PHYSICAL	Physical discomfort due to symptoms (pain, dyspnoea...) and/or injuries (tumorous ulcer...).				
	THERAPEUTICS	Difficulty in adherence to prescriptions and/or access to drugs/techniques.				
**PSYCHO-EMOTIONAL**	PERSONALITY	Psychological vulnerability: rigid personality traits with difficulty adapting to changes (perfectionism, thoroughness, control...), or psychopathology (alcoholism, drug addiction, psychiatric disease, dementia with behavioural disturbance, delirium...).				
	EMOTIONAL	Maladaptive emotional distress (intense, persistent, interfering with relationships and functionality).				
**SPIRITUAL**	SENSE	Deep distress with feelings of rupture due to illness, with difficulty finding meaning in the situation, feelings of incoherence with the actions and decisions taken throughout life.				
	CONNECTION	Deep distress with isolation and rupture of relationships, feelings of guilt, does not feel at peace with others or that they are one of them, difficulty forgiving, inability to use insight.				
	TRANSCENDENCE	Deep distress with difficulty facing everything that will come, in the face of the unknown: panic at dying, at disappearing, the future of those left, difficulty seeing what they will leave, feelings of injustice.				
**SOCIAL AND FAMILY**	RELATIONAL	Relational distress in the family environment that makes patient care difficult.				
	EMOTIONAL	Emotional maladjustment of the caregiver/s (intense, persistent, hindering relationships and functionality) and which makes patient care difficult.				
	PRACTICE	Distress due to difficulty in managing the basic needs of the patient (hygiene, food, safety...).				
	EXTERNAL	Distress due to the lack of effective external support for the cohabiting nucleus.				
	MONEY	Financial distress and/or difficulties in hiring external help and/or accessing resources.				
**ETHICS**	INFORMATION	Distress due to difficulties in the management of information concerning diagnosis and/or prognosis.				
	CLINICAL DECISIONS	Difficulties in clinical decision making (adequacy of diagnostic and/or therapeutic effort).				
	DESIRE TO ADVANCE DEATH	Desire to advance death (DAD) in any degree: thought, intention, decision, plan and/or explicit request.				
**DIRECT RELATIONSHIP WITH DEATH/DYING PROCESS**	LOCATION	Difficulties in planning the place to die (no agreement between patient-caregiver) or request to change location.				
	SITUATION IN THE LAST DAYS (LDS)	Difficulties in managing the dying process (maladaptive denial of the situation, refractory symptoms, difficult sedation).				
	MOURNING	Risk factors during mourning.				
**COMPLEXITY LEVEL:** The highest observed in any of the affected sub-areas:						

**Table 2 ijerph-18-12307-t002:** Key characteristics stratified by gender and age groups *N* (%).

	≤80 Years *N*: 892 (53.2%)			>80 Years *N*: 785 (46.8%)			Total *N*: 1677		
	Women	Men	*p*	Women	Men	*p*	Women	Men	*p*
	326 (36.5%)	566 (63.5%)		410 (52.2%)	375 (47.8%)		736 (43.9%)	941 (56.1%)	
Type ^a^			0.664			0.015			0.000
Cancer	260 (79.8%)	469 (82.9%)		214 (52.2%)	226 (60.3%)		474 (64.4%)	695 (73.9%)	
Organ failure	28 (8.6%)	48 (8.5%)		85 (20.7%)	85 (22.7%)		113 (15.4%)	133 (14.1%)	
Neurologic	16 (4.9%)	23 (4.1%)		26 (6.3%)	11 (2.9%)		42 (5.7%)	34 (3.6%)	
Dementia	14 (4.3%)	16 (2.8%)		65 (15.9%)	40 (10.7%)		79 (10.7%)	56 (6.0%)	
Frailty	8 (2.5%)	10 (1.8%)		20 (4.9%)	13 (3.5%)		28 (3.8%)	23 (2.4%)	
Functional impairment ^b^	267 (81.9%)	431 (76.1%)	0.045	397 (96.8%)	338 (90.1%)	0.000	664 (90.2%)	769 (81.7%)	0.000
Cognitive impairment ^c^	67 (20.6%)	113 (20.0%)	0.833	184 (44.9%)	137 (36.5%)	0.018	251 (34.1%)	250 (26.6%)	0.001
PADES assistance in days ^d^	43 (16–105)	39 (15–93)	0.314	39 (12–106)	31 (11–68)	0.038	41 (14–105)	35 (14–87)	0.077
Caregiver: partner	149 (46.6%)	431 (76.7%)	0.000	34 (8.4%)	169 (45.9%)	0.000	183 (25.2%)	600 (64.5%)	0.000
Professional caregiver	65 (21.2%)	69 (12.9%)	0.002	218 (54.9%)	149 (41.2%)	0.000	283 (40.3%)	218 (24.3%)	0.000

Notes: ^a^ Team agreement on the cause of death in cases with multimorbidity. ^b^ Barthel Index < 100. ^c^ Pfeiffer Test > 2 errors. ^d^ Median and Quartiles 1–3. PADES (Home Care Program, Support Team).

**Table 3 ijerph-18-12307-t003:** Evaluation of Complexity: Includes patients with medium or high complexity. N (%). Gender differences adjusted by age.

		≤80 Years			>80 Years			Total	
	Women	Men	*p*	Women	Men	*p*	Women	Men	*p*
Clinical Complexity	198 (65.6%)	366 (68.2%)	0.442	166 (45.0%)	202 (57.4%)	0.001	364 (54.2%)	568 (63.9%)	0.000
Physical	175 (57.9%)	342 (63.7%)	0.101	154 (41.7%)	180 (51.1%)	0.011	329 (49.0%)	522 (58.7%)	0.000
Therapeutical	130 (43.0%)	203 (37.9%)	0.142	101 (27.4%)	119 (33.8%)	0.061	231 (34.4%)	322 (36.3%)	0.453
Psychological Complexity	156 (52.3%)	297 (56.1%)	0.293	129 (36.3%)	146 (42.0%)	0.127	285 (43.6%)	443 (50.5%)	0.008
Personality	72 (23.9%)	154 (28.7%)	0.132	82 (22.3%)	76 (21.7%)	0.823	154 (23.1%)	230 (25.9%)	0.193
Emotional	129 (42.9%)	257 (47.9%)	0.156	90 (24.5%)	115 (32.8%)	0.015	219 (32.8%)	372 (41.9%)	0.000
Spiritual Complexity	109 (36.5%)	221 (41.8%)	0.133	59 (16.4%)	91 (26.4%)	0.001	168 (25.5%)	312 (35.7%)	0.000
Meaning	82 (27.2%)	169 (31.5%)	0.194	42 (11.4%)	70 (19.9%)	0.002	124 (18.6%)	239 (26.9%)	0.000
Connection	36 (12.0%)	70 (13.1%)	0.646	21 (5.7%)	35 (10.0%)	0.035	57 (8.5%)	105 (11.8%)	0.036
transcendence	74 (24.6%)	167 (31.1%)	0.046	28 (7.6%)	47 (13.4%)	0.012	102 (15.3%)	214 (24.1%)	0.000
Socio-familial Complexity	231 (76.7%)	414 (77.2%)	0.870	230 (62.7%)	246 (70.1%)	0.036	461 (69.0%)	660 (74.4%)	0.019
Relational	71 (23.6%)	128 (23.8%)	0.935	71 (19.3%)	70 (19.9%)	0.840	142 (21.3%)	198 (22.3%)	0.623
Emotional	165 (54.8%)	299 (55.8%)	0.787	125 (34.1%)	142 (40.5%)	0.076	290 (43.4%)	441 (49.7%)	0.014
Practical	137 (45.5%)	260 (48.6%)	0.391	100 (27.2%)	159 (45.3%)	0.000	237 (35.5%)	419 (47.3%)	0.000
External	157 (52.2%)	286 (53.4%)	0.739	170 (46.2%)	186 (53.0%)	0.068	327 (48.9%)	472 (53.2%)	0.090
Financial	59 (19.6%)	109 (20.3%)	0.799	32 (8.7%)	36 (10.3%)	0.489	91 (13.6%)	145 (16.3%)	0.142
Ethical Complexity	84 (28.0%)	149 (28.2%)	0.959	80 (22.3%)	77 (22.5%)	0.942	164 (24.9%)	226 (25.9%)	0.637
Information	32 (10.6%)	63 (11.8%)	0.623	23 (6.3%)	34 (9.7%)	0.088	55 (8.2%)	97 (10.9%)	0.075
STE	61 (20.3%)	96 (17.9%)	0.402	65 (17.7%)	47 (13.4%)	0.111	126 (18.9%)	143 (16.1%)	0.157
DHD	20 (6.6%)	37 (6.9%)	0.887	25 (6.8%)	30 (8.5%)	0.382	45 (6.7%)	67 (7.6%)	0.537
Complexity with Death	230 (76.4%)	431 (80.4%)	0.173	181 (49.3%)	222 (63.2%)	0.000	411 (61.5%)	653 (73.6%)	0.000
Location	190 (63.1%)	356 (66.4%)	0.337	132 (36.0%)	174 (49.6%)	0.000	322 (48.2%)	530 (59.8%)	0.000
SLD	51 (16.9%)	101 (18.8%)	0.494	54 (14.7%)	50 (14.2%)	0.858	105 (15.7%)	151 (17.0%)	0.492
Mourning	151 (50.2%)	238 (44.4%)	0.109	80 (21.8%)	87 (24.8%)	0.343	231 (34.6%)	325 (36.6%)	0.402

Notes: STE, suitability of treatment efforts; DHD, desire to hasten death; SLD, situation during last days of life.

**Table 4 ijerph-18-12307-t004:** Logistic regression analysis to determine the complexity odds ratio by gender, adjusting for age.

	>80 Years		Gender: Man	
	OR (CI95%)	*p*	OR (CI95%)	*p*
Clinical Complexity	0.53 (0.43–0.65)	0.000	1.36 (1.11–1.68)	0.004
Physical	0.56 (0.46–0.69)	0.000	1.36 (1.11–1.67)	0.003
Therapeutical	0.67 (0.54–0.83)	0.000	1.02 (0.82–1.26)	0.862
Psychological Complexity	0.54 (0.44–0.67)	0.000	1.21 (0.98–1.49)	0.070
Personality	0.78 (0.61–0.98)	0.035	1.12 (0.89–1.43)	0.333
Emotional	0.49 (0.39–0.60)	0.000	1.34 (1.08–1.66)	0.008
Spiritual Complexity	0.43 (0.34–0.54)	0.000	1.45 (1.15–1.82)	0.001
Meaning	0.45 (0.35–0.58)	0.000	1.45 (1.13–1.87)	0.003
Connection	0.61 (0.43–0.86)	0.005	1.34 (0.95–1.88)	0.098
transcendence	0.31 (0.23–0.41)	0.000	1.52 (1.16–1.98)	0.002
Socio-familial Complexity	0.60 (0.48–0.75)	0.000	1.21 (0.97–1.52)	0.098
Relational	0.79 (0.62–1.01)	0.057	1.02 (0.80–1.31)	0.846
Emotional	0.49 (0.40–0.60)	0.000	1.16 (0.94–1.43)	0.159
Practical	0.66 (0.54–0.81)	0.000	1.54 (1.25–1.90)	0.000
External	0.89 (0.73–1.09)	0.272	1.17 (0.95–1.43)	0.132
Financial	0.42 (0.31–0.57)	0.000	1.09 (0.82–1.46)	0.544
Ethical Complexity	0.74 (0.58–0.94)	0.012	1.01 (0.80–1.28)	0.931
Information	0.70 (0.49–0.99)	0.046	1.30 (0.91–1.85)	0.144
STE	0.77 (0.59–1.01)	0.061	0.79 (0.61–1.04)	0.092
DHD	1.16 (0.79–1.71)	0.454	1.16 (0.78–1.72)	0.470
Complexity with Death	0.36 (0.29–0.45)	0.000	1.53 (1.22–1.91)	0.000
Location	0.41 (0.34–0.51)	0.000	1.42 (1.15–1.75)	0.001
SLD	0.77 (0.58–1.01)	0.062	1.06 (0.80–1.39)	0.693
Mourning	0.35 (0.28–0.43)	0.000	0.93 (0.75–1.16)	0.520

NOTES: STE, suitability of treatment efforts; DHD, desire to hasten death; SLD, situation during last days of life. OR(CI95%): odds ratio and 95% confidence interval.

## Data Availability

The data presented in this study are available on request from the corresponding author. The data are not publicly available due to IDIAP Jordi Gol not storing personal health data in a public database.

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
