# Peer review of "Gender and Observed Complexity in Palliative Home Care: A Prospective Multicentre Study Using the HexCom Model"

_ijerph, 2021, doi:10.3390/ijerph182312307_

Round 1

Reviewer 1 Report

In general:

 Part of the study entitled “Background “very poorly introduces the reader into the problem studied, contains multiple errors, reveals the lack of experience in using references and in a whole needs re-writing. Part of the study named “Objectives” need corrections, as it already contains the results that have not been presented yet. The conclusion should be more clearly stated. The Authors have difficulties in the proper use of  references. The same data presented in the tables and in the figures

In details:

v.30-31: poorly constructed sentence, women compared here to female mice;sentence starting in v.31: citation is missing; v.34: inappropriate citation; v.35/36 Two names appear (Kohlberg, Gilligan)  without any  support in bibliographical position ; v.39: poor relation between the title of the study  and the problem of empathy; v.41: what does it mean “also” here? no examples of female invisibility has been presented, except of the statement; v.41: inappropriate way of citation (should be: Gott et al.); the sentence starting in v. 43: hard to understand the meaning: 76% had asthenia and only 19% had documentation (may be 19% of women?); besides this is not true, as asthenia is frequently documented symptom in cancer patients; v.52: gender dynamics seems to have inappropriate meaning; In the figure 1.: the “N” is not explained;  v.126: grammar error;  v.147: the  controversial opinion of the Authors that the lower occurrence of cancer in older women can be explained by their age needs profound discussion, bibliographical support etc.) Or may be this is only grammatical error and THUS needs correction;  v.152: are the numbers the percentages? ;v.156: data not confirmed by any reference; v.157: is Etkin a correctly written name (in bibliography there is Etkind). If so, “et al.”is missing and the posion of bibliography is missing; v 157: it seems that this is not appropriate place for this sentence; v.161: hard to agree with the Authors’ conclusion that  acceptance of death has to be related with greater complexity; v.162: hard to understand: in older people lesser pain that in younger? or: lesser pain from cancer than from other diseases? 

Tables: better will be: women, men; it is obvious that “total” means 10%; overall: poorly constructed with editorial errors, this is probably typo error as women are compared with home (men?); Figure 2: the numbers are probably (this is not stated)) the percentages, statistical significance is missing; Figure 2: women , men – will be better; Figure 3: women, men; Figure 3: it would do if the percentages would be  marked only in vertical line; Figure 3: statistical significance is missing

“Objectives” need corrections, e.g. “observed complexity” – the results have not been presented yet, “teams” – this could lead to uncertainty - will it be the study on palliative care teams or patients?

Part of the study “Strengths and limitations” needs some corrections as it stars with limitations what has not been stated .

In “Conclusion”: as the term “complexity” is not generally known, and although explained in the text, should be explained again or may be rather replaced by its described meaning – this is only reviewer’s suggestion.

Reviewer 2 Report

Thank the authors for providing me an opportunity to learn from this manuscript. Overall, the topic is interesting that further links research on palliative home care with gender issues. However, there are still some fundamental issues that I would recommend the authors to address.

(1) Most importantly, I did not see a clear linkage between the issue of androcentrism/ gender inequality and the perspectives the authors presented the findings. This study is more like an exploration regarding female/male patients’ perceptions towards palliative home care and many other associated aspects. There is indeed a difference in terms of gender, but it can be hardly concluded as an “inequality”. The design does not involve obvious measures about the quality of institutions or how female patients were unfairly treated. Therefore, I had some difficulty understanding the authors’ discussion. Shall we just focus on “gender and observed complexity in palliative home care”? Of course, the authors can also start an argument about inequality, but the section of discussion and conclusion must be significantly revised. For instance, how could you judge that some variations/disparities are caused by the presence of “female invisibility” based on this research design/these findings?

(2)After discussion, conclusion, implications and limitations should be placed. It is a bit unintelligible to have a section of “strengths and limitations” in-between discussion and conclusion.

(3) The authors shall provide detailed information about the data collection process. Which sampling method did you adopt? How could you ensure your sample is representative to a target population to some extent? or at least, you have such an awareness.

(4) How a HexCom model operates shall be clearly demonstrated. Moreover, if the authors used a simple Pearson's chi square test to compare differences between female and male patients, the level of significance must be labelled in the tables.

(5) In Table 2, the dichotomy is “women v.s. home”. Is this a typo?

(6)I suggest the authors to revise introduction and research objective too, as the research motivation, considering my comments all above, is not clear.

Major revisions.

Reviewer 3 Report

The topic is interesting and important as it applies HexCom model to investigate the differences in observed complexity based on the gender of the patients of home palliative care, who would be a group of specific population left with little attentions. The overall manuscript is clearly written. However, this article has some concerns to be addressed as follows.

Major concern:

I just wonder whether the statistical analysis of the variables in HexCom model, including clinic, psychoemotional, spiritual, social and family, ethics, direct relationship with death/dying process areas, by using Pearson’s Chi Square test or t-test in table 1, table 2, figure 1, and figure 2, respectively, would be adequate to address the differences between male and female gender and also for the group, more than 80 years or £ 80 years old. I think that the regression analysis of all the significant individual variables would provide more convincible results on the basis of statistically significant.

Minor concern:

  1. I don’t understand why there appear “Home” in the three categorical titles in table 2?
  2. Could the authors mark the statistical significances in Figure 1 and figure 2, that would be more easily for the readers to read and understand.
  3. The arrangements of both the table 1 and table 2 are difficult to read.

Round 2

Reviewer 2 Report

After reading the revised version of the paper and the responses provided by the authors, I want to applaud them for doing an excellent job in addressing the issues raised by me. Now the paper looks more concise and logic. I do believe the issues of gender "inequality" matters to strategic plannings regarding palliative home care services in Spain, and you have revealed empirical evidences in various aspects.

However, since this study is quite grounded, I'd be really grateful if the authors could further provide research implications in the section of conclusions, which is very important in approaching the aim and scope of this study, instead of briefly stating the obtained results. How could a Catalonian decision-maker implement effective reform policies ( I assumes you are unhappy with the present system) to eliminate negative effects of previously ignored “gender issues” in the palliative home care system.

Author Response

Point 1: I'd be really grateful if the authors could further provide research implications in the section of conclusions, which is very important in approaching the aim and scope of this study, instead of briefly stating the obtained results. How could a Catalonian decision-maker implement effective reform policies ( I assumes you are unhappy with the present system) to eliminate negative effects of previously ignored “gender issues” in the palliative home care system.

Response 1: Thank you very much for your comments and invitation to put forward some recommendations. We have re-written the section of "Conclusion" to link our research to some policy recommendations and future lines of research.

Reviewer 3 Report

Accepted.

Author Response

Point 1: Accepted

Response 1: Thank you once again for your valuable comments on previous versions of our manuscript.